# Short-Term Implications of Climate Shocks on Wheat-Based Nutrient Flows: A Global “Nutrition at Risk” Analysis through a Stochastic CGE Model

**DOI:** 10.3390/foods10061414

**Published:** 2021-06-18

**Authors:** Tetsuji Tanaka, Özge Geyik, Bariş Karapinar

**Affiliations:** 1Department of Economics, Setsunan University, 17-8 Ikedanakamachi, Neyagawa, Osaka 572-8508, Japan; 2Centre for Integrative Ecology, School of Life and Environmental Sciences, Deakin University, Burwood, VIC 3125, Australia; ogeyik@deakin.edu.au; 3Center for Economics and Econometrics, Boğaziçi University, Bebek, İstanbul 34342, Turkey; baris.karapinar@boun.edu.tr

**Keywords:** export restrictions, productivity shocks, wheat yields, nutrition security, international trade, computable general equilibrium models

## Abstract

Food security analyses of international trade largely overlook the importance of substantial heterogeneity and complexity of nutrient content in food products. This paper quantifies the extent to which wheat-based nutrient supplies, including energy, protein, iron, zinc, and magnesium, are exposed to the risks of realistic productivity and trade shocks. By employing a static and stochastic world trade computable general equilibrium (CGE) model, we find that productivity shocks may result in losses in households’ nutrient consumption of up to 18% for protein, 33.1% for zinc, and 37.4% for magnesium. Significant losses are observed in countries mostly in the Middle East, North Africa, and Central Asia. Since the main centers of wheat exports have recently been shifting to former Soviet Union countries, we also simulated the nutritional risks of export restrictions imposed by the Russian Federation, Ukraine, and Kazakhstan, which have resorted to this policy instrument in recent years. We find that partial export restrictions increase the probability of nutrient shocks by five times or more in most countries that we studied. Increased nutrient deficiencies have a range of public health implications in the affected countries, which could be mitigated and/or avoided by adjusting production and trade policies and by targeting high nutritional risk groups, such as women and children. Since the potential implications of supply shocks are diffused across countries through international trade, the stricter regulation of export restrictions to enhance the predictably and reliability of global food supplies is also needed.

## 1. Introduction

Wheat is one of the most important sources of nutrients, as it contributes to 19.6% and 18.3% of total human consumption of protein and calories, respectively [1]. It provides other essential macro- and micro- nutrients, such as iron, magnesium, zinc, and folate. For a substantial part of the global population, it is a major source of these nutrients. It is also one of the world’s most traded agricultural products in terms of quantity. In 2019, the global wheat trade amounted to 180 million metric tonnes (Mt) (FAO 2019). International trade supplied almost one-quarter of all wheat-based nutrients at the aggregate level (FAO 2019), hence playing a major role in global food security.

The categorizations of food items in global datasets are broad, and they lack information on nutrient content. As a result, the importance of substantial heterogeneity and complexity of crops’ nutrient content and how the flow of these nutrients through trade might affect nutrition outcomes are largely overlooked.

New challenges to food and nutrition security, beyond hunger, are becoming more prevalent around the world in various forms of macro- and micro-nutrient deficiencies and obesity. An estimated three billion people suffer from at least one form of malnutrition (i.e., hunger, micronutrient deficiencies and overweight/obesity) [2], which has substantial implications for public health. A growing number of studies highlight the importance of enabling governance structures of agri-food systems for positive nutritional outcomes [3,4,5]. Substantial literature also exists on pathways through which international trade of major staple crops may affect food security [6,7,8,9,10,11]. It has an increasingly important role in nutrition security through its impacts on food availability, price and accessibility in addition to effects on consumer demand due to changing food environments [12]. However, the literature on global flows of nutrients contained in traded commodities is still scant [13,14,15,16]. It is important to analyze the nutritional impacts of the wheat industry through international trade because wheat is the most consumed grain as food in terms of calorie intake (FAO, 2019), and a considerable amount of wheat is internationally traded while rice tends to be locally produced and consumed although wheat does not cover all nutritional issues and regions (for example, the deficiency of calcium cannot be improved by more consumption of wheat).

Nutrition security and the international food trade nexus is particularly important in the context of climate change as a risk aggregator, which is affecting both short-term and long-term dietary nutrient supplies. For wheat in particular, studies estimate global yields to decrease by around 6% for every 1 °C temperature increase [17,18,19]. The risk of extreme events is also growing and thus exposing global nutrient supplies to increased frequency and magnitude of short-term shocks, to which certain regions of the world are more susceptible [20,21,22,23,24,25,26]. Through international trade, regions that may not be exposed to yield shocks may become vulnerable to the impacts of shocks occurring elsewhere [27]. Since farming sectors cannot respond quickly in the short term, the impact of these shocks is largely transmitted across countries through trade. Regions that face constraints in domestic supplies may resort to imports in order to mitigate or to avoid nutrient shortages.

Relying on international trade for imports entails its own risks—as importing countries are vulnerable to teleconnected supply shocks [28]. As experienced during and after the food crisis of 2007, net food-importing countries were exposed to supply risks due to exporters’ beggar-thy-neighbor trade policies in the form of export restrictions [8,29]. This is particularly relevant for wheat, since market concentration in exports has been shifting from the Organisation for Economic Co-operation and Development (OECD) countries to former Soviet Union countries that have resorted to imposing restrictions on exports [30]. Market concentration of the top five exporters in 2000—namely, the United States, Canada, France, Argentina, and Australia—has dropped from 79.7% to 48.2% in 2016, while that of the Russian Federation, Ukraine, and Kazakhstan has increased from 4.8% to 22.6% in the same period [1]. The latter group of countries also exhibits a higher degree of variance in total factor productivity (for simplicity we call this “productivity” or “yield” below) (Figure 1). Hence, this geographical and developmental shift in the new centers for wheat exports increases the potential of exporting these countries’ own climatic and political risks to the rest of the world.

In this paper, we contribute to the emerging literature on nutrition security, climate change and international food trade by providing a case study on wheat. We explore the extent to which regional/country connections in nutrient supplies are exposed to risks of realistic productivity shocks and how these shocks affect wheat’s contribution to the daily intake requirement for selected nutrients and energy. We analyze countries’ trade-related risk exposure to short-term wheat supply shocks by adapting a static and stochastic world trade computable general equilibrium (CGE) model. We run simulations by imposing productivity shocks on wheat in line with [26,31]. Employing the CGE model with Monte Carlo draws produces a range of household consumption outputs.

Wheat is frequently regarded not to be a nutrient-rich commodity. Yet, it is a major source of dietary energy, proteins and micronutrients and diversify non-nutrient bioactive food components [32]. However, for the understanding of our results, it is noted that wheat is not the main source of protein in many regions. People in some low-income countries may intake more protein from wheat in comparison with those in high-income countries since affluent households tend to have a variety of options for nutrients. Therefore, impacts estimated could vary depending on the circumstances of each region.

For each region in the model, the paper reports on a risk assessment measure, called value at risk, which is often used in financial risk assessment studies to indicate a set probability for the amount of potential loss in a given time frame. In this paper, we report on wheat consumption at risk at the worst 5% outcome of the Monte Carlo draws for one year (indicating a 1-in-20 years event). Then, these volumes are converted to the corresponding nutrient values, or “Nutrition at Risk (NAR)”, which then is assessed against the recommended daily intake values according to the joint guidelines of the Food and Agriculture Organization (FAO) and the World Health Organization (WHO) [33].

The rest of this study is organized as follows. Section 2 describes the method, data, and scenario designs in this study. Section 3 presents the main findings from this study. Section 4 offers a discussion and policy relevance of these results. Section 5 concludes the article.

## 2. Materials and Methods

We use the global stochastic CGE model extended by [34] based on the single-country CGE model developed by [35], with 2014 global social accounting matrices (SAM) composed of the Global Trade Analysis Project (GTAP) database version 10. The single-country standard model is converted into a stochastic model with the Monte Carlo method in which randomized productivity shocks are generated following the independent and identically distributed (i.i.d.) normal distribution. In total, 25 regions are identified, focusing on wheat-producing, exporting, and importing countries. Table 1 lists the model aggregations that include 25 regions, eight sectors, and four factors of production.

We assume that a representative producer maximizes profit under the Leontief technology for gross output using intermediate inputs and a value-added composite aggregated by production factors with a constant elasticity of substitution (CES) function in which the elasticity of substitution is assumed to be the values from the GTAP database (see Figure 2). Assuming relatively short-term and uncertain situations where farming sectors cannot fully respond to unexpected positive or negative yield shocks, only unskilled labor is mobile across sectors but not internationally. Other factors (skilled labor, capital, farmland, and natural resources) are immobile between sectors and between regions. The primary factors are assumed to be fully employed. Goods produced are allocated between aggregated exports and domestic goods, with a constant elasticity of transformation (CET) function. Domestic goods are combined with aggregated imports to make composite goods, using the CES form as assumed by [36]. Composite imports consist of imports from individual foreign regions, and similarly, composite exports are distributed to various individual countries. Armington elasticities of substitution between the same products from different regions (countries) are typically used in these models to reflect observed preferences for internationally traded products. The elasticity for wheat is assumed to be 4.45 in the GTAP database.

Our model does not explicitly consider different types of wheat. However, the share parameters in the CES/CET functions for international trade were calibrated based on historical trade flows, which is an approximate reflection of the preference of each region. Exchange rates are assumed to be endogenous variables that equate the balance of payments with a net foreign savings variable being exogenous. For domestic investment behavior, the savings-driven closure is adopted to estimate short-term impacts.

The representative household encompasses a two-step consumption structure. At the first stage, food-related goods are substituted and aggregated to produce a food composite with the CES form whose elasticity is assumed to be 0.1 following the estimates for food commodities by [37] (Figure 3). Second, the representative household consumes the synthesized food good and other non-food products and services using a Cobb–Douglas function, which varies the utility level.

### 2.1. Yield Volatility

We basically follow the method of Tanaka and Hosoe (2011) and Erhan et al. (2018) to generate randomized yield shocks. The first step in constructing the stochastic model is to develop exogenous, random productivity shocks for the wheat sector of each region. The wheat productivity for each region is estimated as production quantity divided by areas harvested. Therefore, the productivity or yield considers all the elements relevant to wheat production such as weather, labor, land, fertilizer and machinery. The volatility of wheat productivity is assessed by the following method. The autoregressive integrated moving average (ARIMA) process is fit to time series’ yearly data on wheat yield from the FAOSTAT, which allows us to remove any time trends observed, and the residuals generated from the regressions are used to estimate productivity volatilities. The ARIMA models are expressed as follows:(1)$$Y_{t,r}=\sum_{i=t-p}^{t-1}\delta_{i,r}Y_{i,r}+\sum_{j=t-q}^{t}\theta_{j,r}\mu_{j,r}$$
where $\delta_{i,r}$ and $\theta_{i,r}$ signify the parameters to be estimated, and $Y_{i,r}$ are $\mu_{i,r}$ the first difference of wheat yield and the prediction error in a given period of time, respectively. The subscripts 𝑝, 𝑞, and 𝑟 express the number of autoregressive terms, the number of moving average terms, and the region, respectively. The Akaike information criterion (AIC) is used for model selection and the results and standard deviations of yield volatility are summarized in Table 2.

For each of the selected regions, we generated a set of 1000 Monte Carlo draws to feed into the CGE model. This measures the productivity of wheat as production per acre of harvested area and estimates the standard deviations $\sigma_{r}$ of the productivity of these regions with time series data for 25 years (1992–2016) provided by the FAO. Since our scenarios include the Russian Federation and other former Soviet Union countries, we limited the period to 25 years—as data for these countries are publicly available only after 1992.

We then standardized regional productivity data to that of the reference year of 2014 by dividing all individual annual data by their 2014 productivity level. Next, ARMA models were fitted to filter autocorrelation and non-stationarity. Then, the standard deviations of these residuals ($\sigma_{r}$) were used for generating Monte Carlo productivity shocks for scenario simulations with the function of generating random numbers based on the standard deviations (SD) in Table 2 following the independent and identically distributed normal distribution on the general algebraic modeling system. We simulated 1000 Monte Carlo draws for each scenario. We set up the minimum value of yield variation as 0.3 in order to avoid computational difficulty, which is tolerable and realistic (random shocks only in Morocco exceed the limit 29 in 1000 shocks. However, Morocco is a net importing country, and accordingly the limit does not seriously affect the main conclusion). One thousand Monte Carlo draws were run for each of the 25 regions. Among our 1000 draws, Morocco shows the largest standard deviation of productivity.

The model produces sets of 1000 outputs in the form of household consumption, welfare, prices, export and import quantities, etc., for each scenario. For the purpose of this study, we rely on household wheat consumption outputs indicated as percentage change from the 2014 reference year. We use the “value at risk (VaR)” as a measure of analysis. As stated in the introduction, this measure is often used in financial risk assessment studies to indicate a set probability of the amount of potential loss in a given time frame, in our case one year. Here, we use it in the context of a potential loss of nutrient consumption by setting the value at risk at the worst 5% outcome of our Monte Carlo draws. Hence, we focus on the consumption change percentages that correspond to the worst 5% outcome for each country/region in each scenario. We define this concept as “Nutrition at Risk (NAR)” and report accordingly.

Separately, we calculate the countries’ supply of wheat-based nutrients by multiplying the annual wheat supply for human consumption reported in the FAO’s food balance sheets by unit nutrient values reported in the United States Department of Agriculture’s food composition tables [38]. This provides the values for the annual food supply of protein, calories, magnesium, iron, and folate that are contained in wheat. We also calculate age and gender adjusted annual nutrient requirements for each country’s population by using the recommended nutrient intake values (as suggested by the FAO/WHO). We use 2012 population data from the Population Division of the United Nations [39]. We use the number of births as an estimate of the number of pregnant women to account for their special dietary needs. We also use regional human biomass estimations to calculate protein and energy requirements. We obtain age- and gender-specific recommended intake values of energy [40], protein [41], vitamin and minerals [33]. We assume 10% bioavailability for iron, moderate bioavailability for zinc, and moderate activity for energy requirements. Consequently, the contribution of wheat-based energy, protein, iron, magnesium, and folate to the corresponding annual population requirements is calculated for each country.

### 2.2. Scenarios

We quantify the nutrition consumption impacts on the regional level by conducting comparative stochastic static analyses considering the following scenario factors: (1) wheat yield shocks randomly generated for each region; (2) 50% export quotas on wheat by the former Soviet Union regions (i.e., Russia, Ukraine, and Kazakhstan) (see Table 3).

As no shock is assumed in scenario Reference as a base run, its results are nothing but the original GTAP data. The subsequent scenario is used to investigate the impact of productivity shocks (Y). Scenario YQ is used to analyze the nutrient impacts of the possible export quota impositions of former Soviet Union exporters, namely the Russian Federation, Ukraine, and Kazakhstan under productivity shocks. We assume no intersectoral mobility of capital and simulate export quotas set by the three exporters. The size of the export quota is assumed to be half the original import level from these exporters described in the scenario “Reference”.

## 3. Results

Wheat is a major source of nutrients in the countries studied. We estimate that, on average, populations in 23 countries (Table 4) consumed 67% of recommended protein, 36% of total calories, and 48% of recommended iron from wheat in 2012. Hence, the potential risks of losses of wheat consumption due to yield and trade policy shocks are expected to be significant in relation to the nutrient intake requirements of the populations concerned.

### 3.1. Productivity Shocks on Nutrition

Among our 1000 draws for each country/region, Morocco exhibits the largest standard deviation of land productivity, followed by Nigeria and Australia. Similarly, Iran, Kazakhstan, and Ukraine also show high degrees of volatility. On the other hand, India, Pakistan, and China are the least volatile countries among the 25 countries/regions studied (Figure 1).

We find that extreme shocks, measured here as 1-in-20-year extreme events, lead to a reduction of households’ wheat consumption up to 13.7%. Significant consumption losses are observed in countries mostly in the Middle East, North Africa, and Central Asia. The risk of consumption losses due to productivity change does not seem to depend on whether a country is an exporter or importer of wheat. While consumption in Iran, Morocco, and Egypt—importers of wheat—is susceptible to productivity shocks, Kazakhstan and Ukraine, which are exporters, also exhibit the negative consumption effects of productivity fluctuations. This is because importers are sensitively affected by extreme productivity changes in exporting countries, and exporters also suffer or benefit from bad or good crops in their own regions, respectively.

Reduction in the consumption of wheat leads to various degrees of nutrition losses, as we analyze through NAR values representing the worst 5% outcomes (i.e., 1-in-20-year shock events) as a percentage reduction in gender- and age-adjusted population-level nutritional requirements. For Morocco, NAR values for protein, energy, iron, zinc, and magnesium are 18%, 9.4%, 10.9%, 33.1%, and 37.4%, respectively. For Iran, NAR values for the same set of nutrients are 14.7%, 7.5%, 8.8%, 26.6%, and 29.0%, respectively. NAR for Kazakhstan is −6.3% for protein, −12.3% for zinc, and 13.8% for magnesium (see Table 4). Productivity shocks in wheat seem to affect primarily protein, magnesium, and zinc supplies.

### 3.2. Export Restrictions on Nutrition

Exporting countries that face productivity shocks at home may resort to export restrictions to stabilize/lower domestic prices, hence their domestic consumption. This in turn shrinks exports and aggravates nutritional supply risks abroad, especially in net food-importing countries. As the main centers of gravity in wheat exports have been shifting away from OECD countries to former Soviet Union countries with higher degrees of climatic risk, the latter’s trade policy practices have important implications for the rest of the world. As an additional stressor on wheat supplies, we simulated the nutritional risks of export restrictions imposed by the Russian Federation, Ukraine, and Kazakhstan, which have resorted to this policy instrument in recent years [30].

We find that the probability of wheat-based nutrient losses for households increases substantially across countries when we employ a scenario in which quotas on wheat exports are implemented with productivity shocks (scenario YQ) (Figure 4). When these three countries—the Russian Federation, Ukraine, and Kazakhstan (historically, Russia imposed a export ban on wheat in 2010 due to drought. Ukraine also imposed export tax and quota on wheat during the 2008 food crisis. It is shown that the local prices of wheat in Russia and Ukraine were suppressed by the measures [30]. However, it is demonstrated that an exporting nation that imposes export regulations would economically suffer great losses [34])—enacted export quotas half the size of the amount of their exports in the base year, nutrition shocks that are likely to occur every 50 years or more occur every 20 years in 15 out of 20 countries (after excluding the three countries that impose export restrictions in our scenario) (we use the worst 1% NAR shock value (one in 100 years) under the scenario Y in order to compare with results under the scenario YQ. The worst 1% NAR in the scenario Y is found to be located at around the worst 5% value (one in 20 years) in the scenario YQ). More strikingly, in 12 out of these 20 countries, nutrition shocks that are likely to occur every 100 years or more occur every 20 years. By the same token, we observe that wheat consumption shocks that were likely to happen every other 20 years are now likely to happen approximately every five years in 8 of the 20 countries. In Egypt and Turkey, this probability is as low as every two years. As such, the partial export restrictions imposed by these three exporters increase the nutritional risks substantially in most countries.

In this scenario, Turkey and Egypt are among the list of countries that are likely to experience a 5% loss in daily protein intake from wheat, in addition to Iran, Morocco, and Kazakhstan, which had already been vulnerable to productivity shocks (see Figure 5). The percentage of magnesium loss is highest in Morocco (41.8%) and in Iran (38.6%). Even developed countries, such Australia, France, and Italy, face a decrease in magnesium consumption of close to 5% of their total nutritional requirements (see Table 5). As the export restrictions are employed, international prices of wheat would increase, which reduces domestic consumption of wheat-based nutrients both in importing countries and also in exporting countries that do not impose export restrictions.

As expected, a combination of high per capita wheat consumption and high reliance on imports increases the trade-related risks of policy shocks in the form of export restrictions. High consumption countries such as Morocco, Iran, and Egypt also show high degrees of reliance on imports, as they supply up to 70% of their wheat consumption through imports (see Figure 6). Hence, these countries are more vulnerable to the risks posed by exporters’ adverse trade policies on wheat.

## 4. Discussion and Policy Implications

Although the amount varies from country to country, most countries consume more than the recommended amount of nutrients, which may overshadow the importance of the risks of nutrition losses that may occur due to productivity and policy shocks. Developed countries exhibit the highest degree of overconsumption of nutrients. Per capita wheat-based protein consumption in Italy is the highest among developed countries, followed by France and Spain. Nevertheless, most developing countries display a similar, if not higher, level of nutrient overconsumption, too. For example, per capita wheat-based energy intakes are highest in Morocco and Turkey among the developing countries that we studied. The reduction of the consumption of wheat as a staple food may not pose a significant threat to these countries’ overall food security.

What we observe, however, is the differentiated impacts of weather and policy shocks on nutrients that are contained in a single food crop. While the sources and supplies of each nutrient vary by country, so do nutrient intake requirements. Hence, a supply shock affecting a single food crop may trigger a range of nutritional implications depending on the nutrient composition of the crop, its nutrient contribution to overall nutrient supplies, and the extent of deficiency, if this exists, relating to the nutrients it supplies. On the other hand, because of our focus on a single commodity over the whole household consumption structure, we do not capture the overall change in dietary composition of those affected. Additionally, future analysis with income elasticity of nutrients rather than food per se is needed since nutrient-specific elasticities may reveal different patterns across nutrients [42].

Household consumption impacts within countries are also highly differentiated among income segments [43]. The countries that we find to be exposed to nutritional risks show a moderate degree of income inequality and the prevalence of poverty. In this regard, the extent of consumption changes may not be equal across all households. For example, income shares held by the lowest 20% of households ranged between 5.8% in Turkey to 9.4% in Kazakhstan. The poverty headcount ratio was up to 26.3% of the population in Egypt in 2012 [44]. The risks of nutrition shocks are elevated for low-income household groups in these countries. Since wheat, as a staple food, has a negative income elasticity of demand, lower income households are more exposed to supply shocks than higher income households. While this reflects the quantities of primary food commodities that low-income households consume in these countries, it is also reflective of the quality and diversity of their diets [45]. In the face of a price shock in wheat, lower income households may also reduce the diversity and nutritional quality of their overall diets. However, the model employed in this study provides country-average output without differentiated impacts felt by households from different income levels. For instance, within a given country, poorer households may prefer to invest in the future, such as child education, rather than immediate spending on food [46]. Overall, our results should be interpreted with caution as they do not capture the full spectrum of decision-making dynamics in different within-country contexts.

Deficiencies of the nutrients that we examine affect individuals’ cognitive development, cell growth, immune system, and endocrine system. While the deficiency of protein and magnesium is not common, they are among the most essential nutrients, as the former is essential for the formation of tissues and enzymes and the latter plays important functions in nerve and muscle health and in the immune, skeletal, and cardiovascular systems. Iron deficiency is among the most prevalent micronutrient deficiency, estimated to affect 2 billion people and to contribute to 35 million disability-adjusted life years (DALY) [2,47]. Children and women are particularly at risk. Among the countries we studied, the ones that are particularly vulnerable to nutrition shocks also exhibit high percentages of anemia, which is a commonly used indicator of iron deficiency. In 2012, 28.8% of children and 29.4% of women (of reproductive age) in Kazakhstan, 32.2% of children and 34.2% of women in Morocco, and 30.2% children and 27.9% of women in Iran were reported to have anemia [2]. Similarly, zinc deficiency is estimated to cost 28 million DALY around the world. Almost 5% of the population of Kazakhstan, 19.2% of Morocco, and 23.2% of Iran were estimated to be zinc deficient in 2005 [2]. Hence, these countries would need to design agricultural production and trade policies to mitigate weather- and trade-induced shocks in their overall nutrient supplies and to develop targeted nutrition and health policies for high-risk groups in their societies.

The potential implications of supply shocks are diffused across countries through international trade. Our findings point to an important link between countries’ risk exposure and the combination of three interrelated factors: a country’s own domestic yield volatility, its reliance on wheat imports as the percentage of total supplies per capita, and the share of wheat in its total nutrient supplies (see Figure 7). The higher a country’s own production volatility is and the higher its reliance on imports when it has high share in its food supplies, the greater its risk of loss of wheat-based supplies increases when the rest of the world experiences production volatility and/or exporters impose trade restrictions. Our results suggest that a combination of policy measures to reduce yield volatility (including, for example, the diffusion of cultivars targeted to stabilize yields, extension of irrigation and effective water management practices in rain-fed agricultural systems, and developing widespread adaptation practices to mitigate the impacts of climate change on yields) and the diversification of nutrient supplies in terms of products and of trade partners may effectively decrease future nutrition security risks.

Imports contribute to a large part of wheat-based nutrients in some countries such as Egypt (69%), Morocco (68%), Brazil (49%), Iran (43%), and Italy (35%) [1]. For major importers, the nutrition consumption shocks follow the channel through which yield and/or policy shocks trigger international price hikes, which lower import volumes. This, in turn, results in the loss of household consumption of nutrients. For example, according to our model estimates, Morocco’s 1-in-20 years outcome of loss of protein consumption by 20.1% is derived from a 21.2% increase in the price of its wheat imports due to export supply shocks. While trade connectedness plays an important role in nutrition supplies, it also results in the diffusion of nutritional risks.

Global stability and diversity of supplies, therefore, are important for nutrition security. Export restrictions in the form of quotas or taxes tend to aggravate the short-term impacts of productivity shocks as experienced in the 2007/8 food crisis. While the nutritional gains for the restricting country are minimal, losses on importing countries are substantial. Our model estimates suggest that the Russian Federation’s NAR values are already very low, at 1.9% for protein, 1.1% for energy, and 1.6% for iron. Hence, its restrictions on exports do not provide a significantly positive nutritional outcome. Nonetheless, they lead to substantial increases in the risk of loss of nutrition in many countries such as Morocco, Iran, and Egypt.

While international trade rules largely constrain countries’ import policies, they are lenient in relation to export restrictions [6,48,49]. The World Trade Organization (WTO) rules and regulations seemingly disallow quantitative restrictions, yet they include vaguely defined exceptions that provide ample flexibility for countries to resort to export restrictions without facing significant legal constraints [48]. In addition, WTO rules do not bind export taxes at all, which member countries could set at prohibitively high levels (and hence, effectively functioning as export bans). Notable exceptions in this policy domain are the accession protocols of recently acceded countries to the WTO, including Russia, Ukraine, and Kazakhstan. Their membership obligations on export restrictions are set more strictly than those of founding members of the WTO [49]. Nonetheless, they may still enjoy the availability of exceptions and face limited sanctions if they violate their accession obligations in this policy field. Stricter international regulation of this policy area in order to enhance the predictably and reliability of global food supplies is key to avoiding nutritional shocks [11,50,51].

## 5. Conclusions

Global nutrient supplies are exposed to yield- and policy-induced shocks, which are likely to increase due to climate change. This is particularly important as new challenges to food security, beyond malnutrition and hunger, are multiplying around the world. These new challenges such as macro- and micro-nutrient deficiency and obesity have substantial implications for public health. Through international trade and climate teleconnections, nutritional risks are increasingly interconnected. Hence, in order to mitigate these risks, more emphasis should be placed on policy-relevant research analyzing the heterogeneity and complexity of crops’ nutrient content and the dynamic trade patterns of these nutrients under the impact of productivity variability.

This paper examined the magnitude of the risks associated with consumption loss of wheat-based nutrients, including energy, protein, iron, zinc, and magnesium. For high-risk countries, mainly in the Middle East, North Africa, and Central Asia, governance structures are needed to target vulnerable groups in order to reach positive nutrition outcomes at the national level. A combination of policy measures is necessary to reduce yield volatility and diversify nutrient supplies in terms of products and of trade partners. To enhance nutrition security at the international level, regulators ought to impose a more stringent regulation of export restrictions to enhance the predictably and reliability of global supplies.

It would be worth mentioning the limitations of our research. The CGE model used in this paper captures the variation of household consumption reacting to relative prices and real income, which would not be able to precisely depict consumption behavior. For instance, in developing countries, gendered inequality in intra-household food allocation induces a gender gap in food and nutrition security [52]. In addition, people select foods not only depending on prices but also other factors such as nutrition [53]. These non-price food choice determinants are not considered in the model, which is left for future research.

## Figures and Tables

**Figure 1 foods-10-01414-f001:**
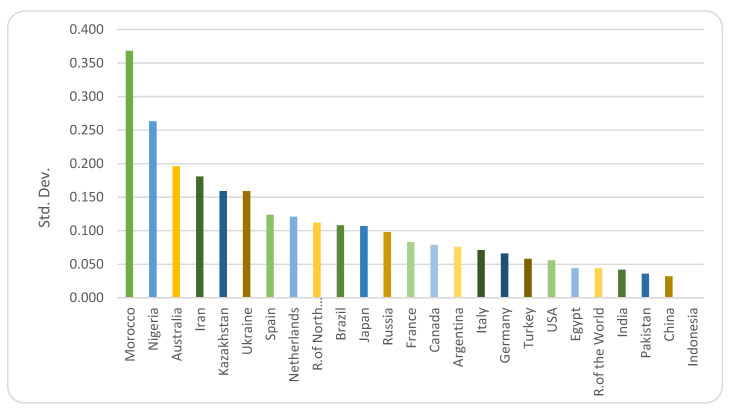
Standard deviations of productivity (ARIMA residuals) in 25 countries/regions, 1992–2016.

**Figure 2 foods-10-01414-f002:**
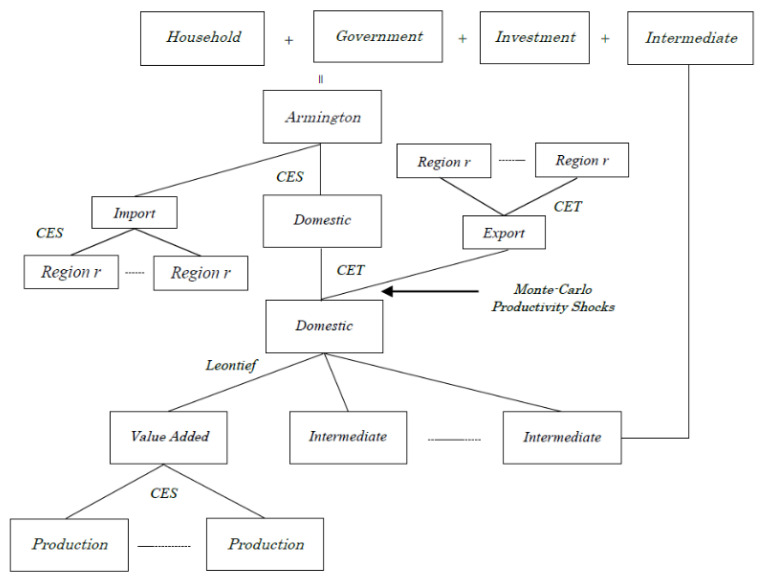
Overview of the model.

**Figure 3 foods-10-01414-f003:**
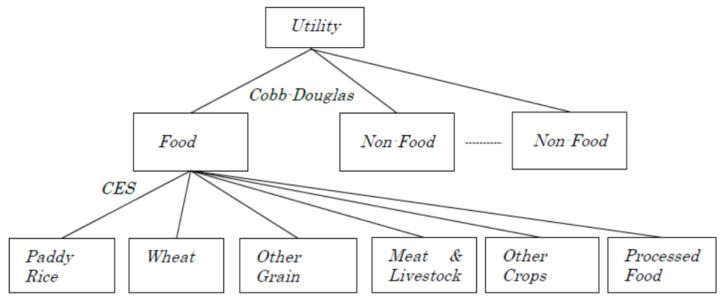
Structure of household consumption.

**Figure 4 foods-10-01414-f004:**
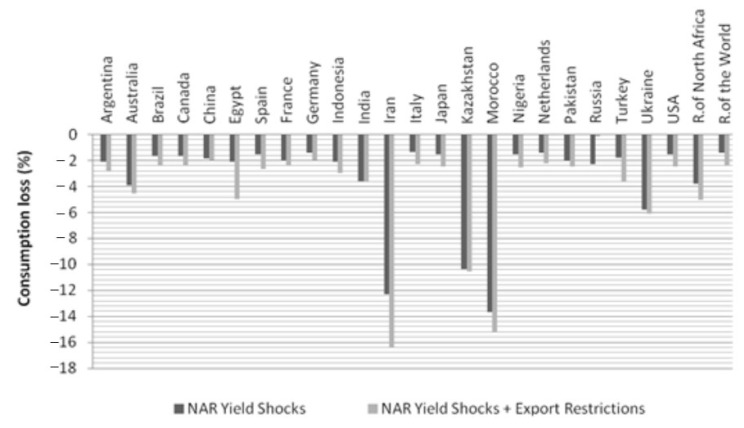
Wheat consumption at risk due to productivity shocks + export restrictions.

**Figure 5 foods-10-01414-f005:**
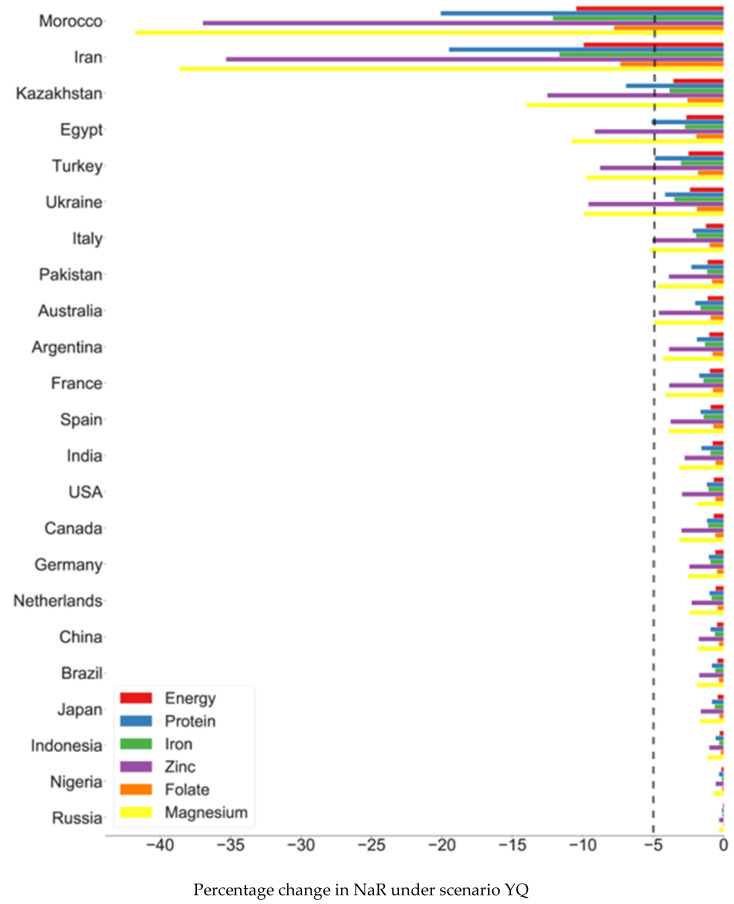
Loss in nutrients (NAR in percentage of daily intake requirements) under the yield shocks + export restrictions scenario (Scenario YQ). Dashed lines highlight the losses above 5% threshold.

**Figure 6 foods-10-01414-f006:**
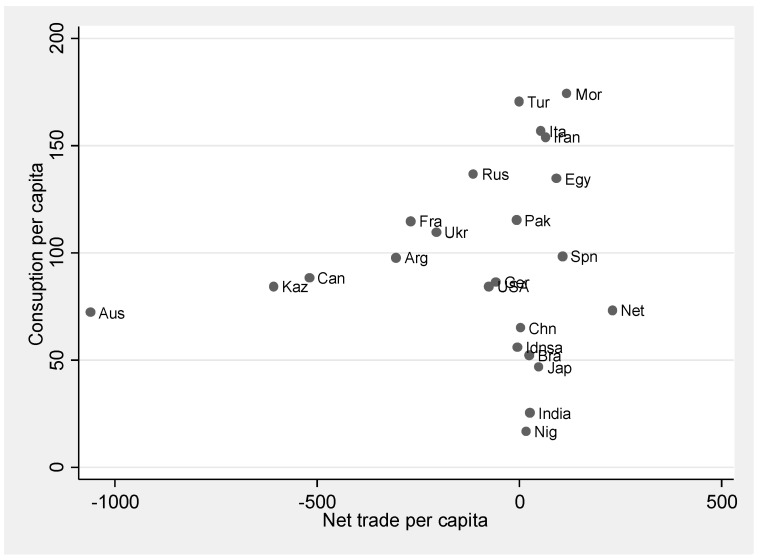
Production versus imports (per capita) of wheat, 2016. Scheme 2019.

**Figure 7 foods-10-01414-f007:**
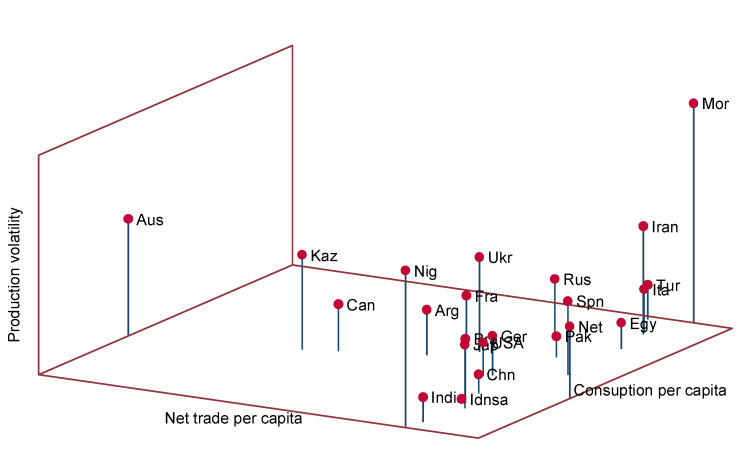
Production volatility, consumption and trade per capita, 2012.

**Table 1 foods-10-01414-t001:** Regional, sectoral and factor aggregations in the model.

Regions		Sectors	Factors of Production
Argentina	Kazakhstan	Paddy rice	Labor
Australia	Morocco	Wheat	Capital
Brazil	Netherlands	Other cereals	Farmland
Canada	Nigeria	Other crops	Natural resources
China	Pakistan	Livestock	
Egypt	Russia	Food	
France	Spain	Transport	
Germany	Turkey	Others	
India	Ukraine		
Indonesia	USA		
Iran	Rest of North Africa		
Italy	Rest of the World		
Japan			

**Table 2 foods-10-01414-t002:** Results of the autoregressive integrated moving average (ARIMA) model.

	Autoregressive Factor			Moving Average Factor			SD
	$\delta_{1}$	$\delta_{2}$	$\delta_{3}$	$\theta_{1}$		$\theta_{2}$	$\theta_{3}$
Argentina	0.14	−0.47		−1.00			0.076
Australia	0.06	0.13		−1.00			0.196
Brazil	−0.26			−1.00			0.108
Canada	−0.42			0.00			0.079
China	−0.44						0.032
Egypt	−0.36	−0.19					0.044
Spain	−0.41			−1.00			0.132
France	−0.34			−0.99			0.083
Germany	0.19	−0.34		−1.00			0.066
India	0.23						0.042
Indonesia	N/A			N/A			N/A
Iran (Islamic Republic of)	0.54			−1.00			0.181
Italy	−0.96			0.77			0.071
Japan	0.73			−1.24	−0.49	0.74	0.107
Kazakhstan	−0.28			−1.00			0.159
Morocco	−0.51	0.11		−1.00			0.368
Nigeria	−1.77	−0.79		1.89	0.95		0.263
Netherland	0.28			−1.76	0.99		0.121
Pakistan	−0.21			−1.00			0.036
Russian Federation	0.87	−0.38		−1.98	0.99		0.098
Turkey	−0.28			−0.72			0.058
Ukraine	−0.34						0.159
United States of America	0.18	−0.39		−1.00			0.061
Rest of North Africa	1.15	−0.71	−0.15	−2.10	2.10	−0.99	0.112
Rest of the World	0.03			0.99			0.044

Note: The model was not run for Indonesia since the country does not produce wheat.

**Table 3 foods-10-01414-t003:** Scenario table.

	Scenario Factor	
	Yield Shock	Export Quota
Reference		
Y	Yes	
YQ	Yes	Yes

**Table 4 foods-10-01414-t004:** Nutrition at risk (NAR) values, productivity shocks (Scenario Y) *.

	Protein	Energy	Iron	Zinc	Folate	Magnesium
Argentina	−1.4%	−0.8%	−1.0%	−2.9%	−0.6%	−3.2%
Australia	−1.8%	−1.0%	−1.4%	−4.0%	−0.8%	−4.3%
Brazil	−0.6%	−0.3%	−0.4%	−1.2%	−0.3%	−1.3%
Canada	−0.8%	−0.5%	−0.8%	−2.0%	−0.4%	−2.1%
China	−0.9%	−0.5%	−0.6%	−1.7%	−0.3%	−1.8%
Egypt	−2.2%	−1.1%	−1.2%	−3.8%	−0.8%	−4.5%
Spain	−0.9%	−0.5%	−0.8%	−2.1%	−0.4%	−2.2%
France	−1.4%	−0.8%	−1.2%	−3.2%	−0.6%	−3.4%
Germany	−0.8%	−0.4%	−0.7%	−1.7%	−0.3%	−1.8%
India	−0.4%	−0.2%	−0.3%	−0.7%	−0.2%	−0.8%
Indonesia	−1.6%	−0.8%	−1.0%	−2.8%	−0.6%	−3.2%
Iran	−14.7%	−7.5%	−8.8%	−26.6%	−5.5%	−29.0%
Italy	−1.3%	−0.8%	−1.2%	−3.1%	−0.6%	−3.2%
Japan	−0.5%	−0.3%	−0.4%	−1.0%	−0.2%	−1.1%
Kazakhstan	−6.8%	−3.5%	−3.8%	−12.3%	−2.6%	−13.8%
Morocco	−18.0%	−9.4%	−10.9%	−33.1%	−7.0%	−37.4%
Nigeria	−0.2%	−0.1%	−0.1%	−0.4%	−0.1%	−0.5%
Netherlands	−0.6%	−0.4%	−0.6%	−1.4%	−0.3%	−1.5%
Pakistan	−1.9%	−1.0%	−1.0%	−3.2%	−0.7%	−3.9%
Russia	−1.9%	−1.1%	−1.6%	−4.4%	−0.9%	−4.6%
Turkey	−2.4%	−1.2%	−1.5%	−4.3%	−0.9%	−4.8%
Ukraine	−4.0%	−2.3%	−3.3%	−9.1%	−1.8%	−9.4%
USA	−0.8%	−0.5%	−0.7%	−1.8%	−0.4%	−1.2%

* Losses above 5% of daily intake requirements are highlighted.

**Table 5 foods-10-01414-t005:** NAR values, productivity & export restriction-induced shocks (Scenario YQ) *.

	Protein	Energy	Iron	Zinc	Folate	Magnesium
Argentina	−1.9%	−1.0%	−1.4%	−3.9%	−0.8%	−4.3%
Australia	−2.0%	−1.2%	−1.7%	−4.6%	−0.9%	−4.9%
Brazil	−0.9%	−0.5%	−0.6%	−1.8%	−0.4%	−1.9%
Canada	−1.2%	−0.7%	−1.1%	−3.0%	−0.6%	−3.2%
China	−1.0%	−0.5%	−0.6%	−1.8%	−0.4%	−1.9%
Egypt	−5.1%	−2.7%	−2.8%	−9.2%	−2.0%	−10.8%
Spain	−1.7%	−1.0%	−1.4%	−3.8%	−0.8%	−3.9%
France	−1.8%	−1.0%	−1.5%	−3.9%	−0.8%	−4.1%
Germany	−1.1%	−0.6%	−1.0%	−2.5%	−0.5%	−2.5%
Indonesia	−0.6%	−0.3%	−0.4%	−1.0%	−0.2%	−1.2%
India	−1.6%	−0.8%	−1.0%	−2.8%	−0.6%	−3.2%
Iran	−19.5%	−10.0%	−11.7%	−35.3%	−7.4%	−38.6%
Italy	−2.2%	−1.3%	−2.0%	−5.1%	−1.0%	−5.2%
Japan	−0.9%	−0.5%	−0.7%	−1.6%	−0.3%	−1.7%
Kazakhstan	−7.0%	−3.6%	−3.9%	−12.5%	−2.6%	−14.0%
Morocco	−20.1%	−10.5%	−12.1%	−37.0%	−7.8%	−41.8%
Nigeria	−0.3%	−0.2%	−0.2%	−0.6%	−0.1%	−0.7%
Netherlands	−1.0%	−0.6%	−0.9%	−2.3%	−0.5%	−2.4%
Pakistan	−2.3%	−1.2%	−1.2%	−3.9%	−0.9%	−4.7%
Russia	−0.1%	−0.1%	−0.1%	−0.3%	−0.1%	−0.4%
Turkey	−4.9%	−2.5%	−3.1%	−8.8%	−1.8%	−9.8%
Ukraine	−4.2%	−2.4%	−3.5%	−9.6%	−1.9%	−9.9%
USA	−1.2%	−0.7%	−1.1%	−3.0%	−0.6%	−1.9%

* Losses above 5% of daily intake requirements are highlighted.

## Data Availability

The data generated in this research will be available upon request.

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
