# Peer review of "Short-Term Implications of Climate Shocks on Wheat-Based Nutrient Flows: A Global “Nutrition at Risk” Analysis through a Stochastic CGE Model"

_foods, 2021, doi:10.3390/foods10061414_

Round 1

Reviewer 1 Report

Dear Authors,

thanks for replying satisfactorily to the points raised in my revision.

The only point you did not fully addressed is the one related to the nutritional decisions. Your text is now correct in highlighting the inter-households  inequality which may bias the data. I think you should also add a reflection related to intra-household inequality. The main point is to warn the reader that results should be taken with some caution as these big global CGE models might not be fully suited to capture nutritional individual decision and can indicate directions but certainly not  detailed and precise results.

Reviewer 2 Report

The is a well-written paper, however, the research question and design are flawed leading to misleading results. I will elaborate as follows:

  1. Line 52-54 on page 2, the authors stated that “nutrition security often ignores the importance of international trade.” This statement ignores that there is a significant amount of literature using different CGE models to study the links between diets, environmental impacts and trade.

A: Thank you for your feedback on this statement. We have revised the respective paragraph for clarification:

“New challenges to food and nutrition security, beyond hunger, are becoming more prevalent around the world in various forms of macro- and micro- nutrient deficiencies and obesity. An estimated three billion people suffer from at least one form of malnutrition (i.e., hunger, micronutrient deficiencies and overweight/obesity) [2], which has substantial implications for public health. A growing number of studies highlight the importance of enabling governance structures of agri-food systems for positive nutritional outcomes [3-5]. Substantial literature also exists on pathways through which international trade of major staple crops may affect food security [6-11]. It has an increasingly important role in nutrition security through its impacts on food availability, price and accessibility in addition to effects on consumer demand due to changing food environments [12]. How-ever, the literature on global flows of nutrients contained in traded commodities is still scant [17-20].

R: There is no doubt that nutrition security is important. This paper, however, only focuses on wheat security. It is necessary to discuss the importance of wheat security rather than nutrition security in general since the two securities are very different. For example, people in South Asian countries lack calcium, which cannot be improved by having more wheat.

  1. There is an inconsistency between research question and answer: the research gap between nutrition security and international trade is a huge one and this paper actually only studied the wheat security and international trade.

A: Thank you for pointing out that our research question was not clear enough. We agree that the relation between international trade and nutrition security is a complex one and there is a wide range of causal pathways between the two. In order to clarify our contribution, we revised as follows:

“In this paper, we contribute to the emerging literature on nutrition security, climate change and international food trade by providing a case study on wheat. We explore the extent to which regional/country connections in nutrient supplies are exposed to risks of realistic productivity shocks and how these shocks affect wheat’s contribution to the daily intake requirement for selected nutrients and energy. We analyze countries’ trade-related risk exposure to short-term wheat supply shocks by adapting a static and stochastic world trade Computable General Equilibrium (CGE) model. We run simulations by imposing productivity shocks on wheat in line with [31, 36]. Employing the CGE model with Monte Carlo draws produces a range of household consumption outputs.”

R: The underlying assumption of choosing wheat as a case study for the research of nutrition security, climate change and international food trade is that wheat is: wheat is a representative nutrition security issue. It is noticed that there are geographical and cultural differences in terms of staple food and wheat is not a primary choice as staple food in some countries and regions, e.g. Thailand. Although ‘nutrition security’ is used here, the discussion around wheat supply and demand actually lies in ‘food security’. The two terms have different meanings. Please find it in the reference here: https://www.nature.com/articles/s43016-019-0002-4. The discussions over what nutrients are deficient, how to meet this demand and the role of wheat in this nutrition security.

  1. Wheat security is only one small part of nutrition security. There are more challenges, such as double-burden leading to non-communicable diseases etc., not been addressed. In this context, nutrition security concerns the overconsumption of energy, salt, sugar, saturated fat while the underconsumption of fibre, vegetables, fruits, micronutrients. Wheat as a source of energy is not a major problem in this regard. From nutritional perspective, increasing meat consumption is regarded as an efficient way to improve protein intake. People in some developing countries may need to increase the consumption of meat to improve nutrition security. This is not considered in the analysis. The authors failed to see the bigger picture of nutrition security, especially wheat is not the main source of protein in many countries.

A: Thank you for your insightful feedback. We acknowledge the overarching concept of triple burden of malnutrition and the complexity of nutrition security. Although wheat is not the richest source of many micronutrients, it still provides between 35% - 60% of their recommended intake in 23 countries included in Nutrition at Risk analysis. This is of course a result of current production and supply patterns. However, although we provide a broader background and discussion for the nexus of nutrition and climate- and trade-related shocks, our assessment focuses on changes in household wheat consumption as a result of those shocks. Therefore, exposure to shock analysis of other commodities or closing dietary nutrient gaps are out of our scope.

Nevertheless, we believe that our revisions in Introduction and Discussion as per the suggestions by Reviewer 1 and 2 have improved the background picture and limits of our study.

R: I have to emphasise that nutrition security is not equivalent to wheat security. The answer above is still misleading to readers. The revision has not clarified the scope of research and the context where the simulations are built clearly.

  1. The wheat yield is influenced by weather factors and the ARIMA on page 5 is trying to remove the weather shocks. The residues in the ARIMA models, however, include not only volatilities of weather but also the other factors, such as government policy changes, pests and plant diseases, labour productivity changes etc. Attributing all the factors to weather volatilities results in biases.

A: Thank you for the indication, which is correct. We modified all the relevant expressions in the paper, removing ‘weather’ or replacing with ‘productivity.’

R: Removing weather volatilities is a technical problem which requires revisions in your methodologies. If ‘weather’ is replaced by ‘productivity’, this means that the paper is not writing the methods actually used. This raises an even bigger technical concern.

  1. GTAP data’s most recent reference year is 2014 and the authors used the older reference year 2011. This contradicts to common sense since the newer data will improve the performance of prediction. Why the authors didn’t use it?

A: Yes, 2014 is the latest version of the GTAP database. We checked the reference year, and found that the year of the database was 2014 so 2011 was changed to 2014.

R: OK.

  1. Line 223-237 on page 7 do not seem to belong to the paper.

  1. The results in Line 239-241 on page 7 are not consistent with nutritional and public health studies.

A: Your comment is appreciated. We removed the paragraphs irrelevant to this paper.

R: OK.

Round 2

Reviewer 2 Report

Thanks for the revisions. My biggest concern for this paper is: yield volatility is the core of the model simulation, however, it is not clear how these volatilities are generated. Did the authors actually considered weather shocks? What other factors did they include? Without a clear explanation of how the simulations were developed, it is hard to make sense of the results. 

Author Response

Responses to the comments

Thank you for your comments. I added explanations about how we generated the randomized Monte-Carlo shocks and what are considered in the productivity as follows. In the manuscript, the sentences revised are in green.

Round 3

Reviewer 2 Report

Thanks for your revisions. The paper is clearer now.

This manuscript is a resubmission of an earlier submission. The following is a list of the peer review reports and author responses from that submission.

Round 1

Reviewer 1 Report

The is a well-written paper, however, the research question and design are flawed leading to misleading results. I will elaborate as follows:

  1. Line 52-54 on page 2, the authors stated that “nutrition security often ignores the importance of international trade.” This statement ignores that there is a significant amount of literature using different CGE models to study the links between diets, environmental impacts and trade.
  2. There is an inconsistency between research question and answer: the research gap between nutrition security and international trade is a huge one and this paper actually only studied the wheat security and international trade.
  3. Wheat security is only one small part of nutrition security. There are more challenges, such as double-burden leading to non-communicable diseases etc., not been addressed. In this context, nutrition security concerns the overconsumption of energy, salt, sugar, saturated fat while the underconsumption of fibre, vegetables, fruits, micronutrients. Wheat as a source of energy is not a major problem in this regard. From nutritional perspective, increasing meat consumption is regarded as an efficient way to improve protein intake. People in some developing countries may need to increase the consumption of meat to improve nutrition security. This is not considered in the analysis. The authors failed to see the bigger picture of nutrition security, especially wheat is not the main source of protein in many countries.
  4. The wheat yield is influenced by weather factors and the ARIMA on page 5 is trying to remove the weather shocks. The residues in the ARIMA models, however, include not only volatilities of weather but also the other factors, such as government policy changes, pests and plant diseases, labour productivity changes etc. Attributing all the factors to weather volatilities results in biases.
  5. GTAP data’s most recent reference year is 2014 and the authors used the older reference year 2011. This contradicts to common sense since the newer data will improve the performance of prediction. Why the authors didn’t use it?
  6. Line 223-237 on page 7 do not seem to belong to the paper.
  7. The results in Line 239-241 on page 7 are not consistent with nutritional and public health studies.

Reviewer 2 Report

The paper analyses the effect of random yield shocks on wheat (combined with export restriction measures) on global nutrition effects employing a global CGE model.

Please, find below my detailed comments.

Introduction:

I would suggest to include in the literature review on trade and climate change the two following papers:

  • Janssens, C., Havlík, P., Krisztin, T. et al. Global hunger and climate change adaptation through international trade. Nat. Clim. Chang. 10, 829–835 (2020). https://doi.org/10.1038/s41558-020-0847-4
  • Nechifor, V., Ferrari, E. Trading for climate resilience. Nat. Clim. Chang. 10, 804–805 (2020). https://doi.org/10.1038/s41558-020-0875-0

Materials and Methods

The paper refers to the nutrient flows of trade but there is no reference to nutrients in the model itself. All the calculations of nutrients are based on an ex-post based. This looks like a shortcoming of the model as consumers are making consumption decision on the basis of their utilities but not on the basis of their nutritional status or requirements. This should be clearly identified, possibly elaborating on how this affect the results of the simulations, by the authors and possibly suggesting future way to overcome this shortcomings.

Line 173: what does it mean setting up the minimum (minimum or maximum ?) value of yield variation as 0.3? How many exogenous observations where beyond this limit?

Line 215: As no shock is assumed in scenario Y, while from table 3 the scenario Y looks like the yield shock scenario.

Lines 223-237 where not delayed from the template.

Results

Line 256-260: authors should elaborate more on these results and try to explain why is that so.

Line 265-267: reported figures should be preceded by a minus in the text or clearly stating they refer to a reduction.

Line 286-289: the concept of shocks occurring every x years should be better clarified.

Section 3.2 should also report if the measures taken by the exporting countries are effective to improve internal food security in those countries applying them.

There is no introduction of the YQ scenario and it is not clear when the authors stop describing scenario Q and move to the next scenario. Section 3.2 looks also like including some repetition (table 4 and some text is repeted).

All in all Section 3.2 and 3.3 should be rewritten as they are not clear and reader is confused by repeating tables and text.

Discussion and Policy Implications

The authors should also report on caveats of the study. For instance they never refer to the fact that nutritional decision are made at individual level while the data and results reported are at country level, implying that they should be considered as average not taking into account not only country inequality but also intra-households inequality